# DP-OPT: Make Large Language Model Your Privacy-Preserving Prompt Engineer

**Junyuan Hong[1], Jiachen T. Wang[2], Chenhui Zhang[3], Zhangheng Li[1], Bo Li[4], Zhangyang Wang[1]**
[1]University of Texas at Austin, [2]Princeton University, [3]MIT, [4]University of Chicago
`{jyhong,zoharli,atlaswang}@utexas.edu, tianhaowang@princeton.edu`
`chenhui5@mit.edu, bol@uchicago.edu`

## Abstract

Large Language Models (LLMs) have emerged as dominant tools for various tasks, particularly when tailored for a specific target by prompt tuning. Nevertheless, concerns surrounding data privacy present obstacles due to the tuned prompts' dependency on sensitive private information. A practical solution is to host a local LLM and optimize a soft prompt privately using data. Yet, hosting a local model becomes problematic when model ownership is protected. Alternative methods, like sending data to the model's provider for training, intensify these privacy issues facing an untrusted provider. In this paper, we present a novel solution called *Differentially-Private Offsite Prompt Tuning* (**DP-OPT**) to address this challenge. Our approach involves tuning a discrete prompt on the client side and then applying it to the desired cloud models. We demonstrate that prompts suggested by LLMs themselves can be transferred without compromising performance significantly. To ensure that the prompts do not leak private information, we introduce the first private prompt generation mechanism, by a differentially-private (DP) ensemble of in-context learning with private demonstrations. With DP-OPT, generating privacy-preserving prompts by Vicuna-7b can yield competitive performance compared to non-private in-context learning on GPT3.5 or local private prompt tuning. Codes are available at `https://github.com/VITA-Group/DP-OPT`.

## 1 Introduction

When Large Language Models gain vast knowledge and versatile ability from large-scale pre-training, prompt engineering has surfaced as the most effective, cost-efficient, and adaptable method to tailor LLMs for a range of downstream applications. In contrast to the resource-heavy optimization of model parameters, prompt engineering merely necessitates API access and iteratively refines prompts based on the validation of training instances. Though manual prompt engineering has achieved impressive performance in various tasks (Petroni et al., 2019; Zhou et al., 2022), it often requires decent human experience in prompt designing and domain knowledge for downstream tasks, including legal judgement (Trautmann et al., 2022), healthcare (Wang et al., 2023b) and art (Oppenlaender et al., 2023). To mitigate the high costs, data-driven prompt tuning was proposed to automate the process. The most prominent example of this is soft prompt tuning, where prompts are characterized as trainable embedding vectors and are refined using a collection of training instances (Houlsby et al., 2019; Roberts et al., 2019; Brown et al., 2020; Chen et al., 2022).

However, one major barrier to the applications of prompt tuning is data privacy. When searching for a validate prompt for an LLM API, such as ChatGPT, there is a need to upload a multitude of training samples for evaluation queries. In privacy-sensitive scenarios, the operation could be prohibited due to two concerns. 1) *Data Confidentiality*. Certain data, like medical histories, proprietary system logs, and personal messages, are inherently confidential and should not be transmitted beyond local devices, internal computing systems, and mobile phones. 2) *Information Leakage*. Resources derived from private data might inadvertently contain personally identifiable information. For instance, personal identifiers like names, residential addresses, and contact numbers present in the fine-tuning data (Lukas et al., 2023) or during the pre-training phase (Carlini et al., 2021) might be retrievable from the adjusted parameters. Even with a limited parameter set, such as prompts, the potential for data breaches remains significant (Duan et al., 2023a).

Figure 1: Differentially-Private Offsite Prompt Tuning (DP-OPT) works as an intermediate layer between local data and cloud models. Leveraging a local model, DP-OPT can fine-tune a differentially-private prompt that can transfer to the target model.

A straightforward approach would be to manage the entire prompt process on the local device and offer services via an API. However, this becomes impractical when there's a preference for a sophisticated closed-source model, not to mention the substantial costs involved in hosting and overseeing an LLM locally. For example, there is a high demand for serving prompts with the most powerful LLMs, e.g., GPT-3.5, to leverage the state-of-the-art generation ability. Yet, the specifics and structure of GPT-3.5 remain proprietary and undisclosed for protecting Intelligent Property (IP). Even if GPT proprietors are willing to support local prompt tuning by dispatching compressed models (Xiao et al., 2023), they could be confronted with the potential peril of losing their model's ownership.

In this paper, we propose Differentially-Private Offsite Prompt Tuning (DP-OPT) to make LLM engineer private and transferable prompts for cloud-hosted LLMs, which is illustrated in Fig. 1. The crux of privacy protection is that DP-OPT operates exclusively on the client. Given a confidential training dataset, DP-OPT uses a few samples as demonstrations to guide a local LLM to generate prompts. The local assistant LLM may be significantly smaller than the intended cloud-based LLMs. Such a prompt generation process is facilitated by a Differentially-Private (DP) ensemble of in-context learning with disjoint private demonstration subsets. Our contributions are summarized as follows:

- We proposed the first end-to-end framework where DP-OPT operates on client devices and data and yields a front-end prompt for inference on the privacy-untrusted cloud. Our framework is the first solution that simultaneously protects (i) data confidentiality by keeping data local; (ii) information privacy by the DP noise mechanism; and (iii) cloud model ownership and IP by eliminating the parameter exposure of cloud models from local training.

- We first show that discrete prompts automatically tuned by LLMs are transferable across models with favorable performance on cloud models. The finding motivates the offsite prompt tuning (OPT) framework with local prompt tuning and cloud inference.

- We then provide the first Differentially-Private mechanism for generating private prompts without gradients or public data. The privacy costs can be tightly bounded during prompt tuning.

- Empirically, our method presents an outstanding performance on multiple language tasks. Prompts tuned on open-source Vicuna-7b (Chiang et al., 2023) can achieve significant performance gains across 4 tasks after transfer to closed-source heterogeneous-architecture models (GPT3.5) or open-source models (Llama-2 (Touvron et al., 2023) or Vicuna-33b).

## 2 RELATED WORK

**Discrete Prompt Tuning.** On the rise of pre-trained generative language models, discrete prompt tuning (Shin et al., 2020) was introduced to amplify the inference capability of LLMs through demonstrations (Brown et al., 2020) or informative instructions (Prasad et al., 2022; Shin et al., 2020). The method is orthogonal to the soft prompt tuning (Lester et al., 2021; Liu et al., 2021) that optimizes continuous prepended embeddings instead of discrete tokens and is therefore not favored for API-only models. One line of the discrete prompt tuning aims to improve the search efficiency by gradient-free phrase editing (Prasad et al., 2022), reinforcement learning (Deng et al., 2022), embedding optimization (Shi et al., 2022), and projected-gradient optimization (Wen et al., 2023). Another line of work does not rely on search or optimization algorithms and was initiated by Automatic Prompt Engineering (APE) (Zhou et al., 2022) that employed LLM to prompt themselves. Later, Deep Language Network (DLN) (Sordoni et al., 2023) improves APE by backward updates and stacked language models. Though these methods achieve great progress in discrete prompt tuning approaching the soft prompt results, the privacy risks or protection of the generated prompts have not been studied yet. In our work, we first highlight the advantage of DLN prompts in transfer learning.

Essentially different from the negative transfer effect presented in recent work (Wen et al., 2023), we show that DLN prompts can transfer to and work better on larger models than on the source models, namely positive transfer. Meanwhile, we provide the first solution to ensure the privacy of the gradient-free algorithms that demonstrate strong empirical performance compared to in-context learning and previous private gradient-based competitors.

**Privacy Risks in Prompt Tuning.** LLMs have been shown to possess the ability to memorize data, not just from extensive pre-training datasets (Carlini et al., 2021; 2022b; Wang et al., 2023a), but also from more concise private prompt-tuning datasets (Duan et al., 2023a). Consequently, it becomes imperative to shield private data from potential exposure in released prompts. To illustrate this vulnerability, Duan et al. (2023a) employed a membership-inference attack (Shokri et al., 2017; Carlini et al., 2022a) to probe the susceptibilities of tuned soft prompts, revealing a significant success rate for the attacks. Though the risk of soft prompt tuning has been highlighted, it is unclear how the discrete prompts exemplified by DLN will leak private information. Our research uncovers direct data leakage via prompts crafted by DLN, underscoring the need for novel discrete defense mechanisms.

**Mitigating Privacy Leakage in Prompt Tuning.** As a golden standard for bounding privacy risks, there is increasing interest to incorporate differential privacy (DP) (Dwork, 2006) into prompt tuning for privacy protection. Closely related to our work, PromptPATE utilized a DP ensemble approach to label public data. Using these as in-context examples, they devised a discrete prompt tailored for few-shot learning on designated models (Duan et al., 2023a). However, the work assumes a set of non-private data which may not hold in practice. In the absence of public datasets, Duan et al. (2023a) also showed the viability of DP-SGD (Abadi et al., 2016) in the realm of soft prompt tuning. In parallel, DP In-Context Learning (ICL) (Wu et al., 2023) advocated for ensembling multiple in-context samples to predict classification labels directly, which limits the query times by the number of available private samples. Instead of prediction, Tang et al. (2023) proposed ensemble generation by prompting LLMs with examples. The generated samples can be used for ICL and enable infinitely many queries. Yet, these approaches were used with the full exposure of private samples to the target model, rendering it infeasible when a model vendor is untrustworthy. From a technical standpoint, our strategy differs from these methods on ensembling outputs to produce instructional prompts. Specifically, we ensemble in-context examples to produce instructional prompts rather than labels. This design facilitates the effortless migration of trained prompts to many models, eliminating the need for extra training on cloud models.

Another focal point among the community is the sanitization of texts (Feyisetan et al., 2020; Xu et al., 2020; Carvalho et al., 2023; Du et al., 2023; Utpala et al., 2023). These works introduce randomness at the word level and are able to achieve the privacy guarantee in terms of local differential privacy (LDP) (Kasiviswanathan et al., 2011) or a similar definition called *metric differential privacy*. Recent advancements in this field (Mattern et al., 2022; Utpala et al., 2023) combine the idea of perturbation with paraphrasing based on fine-tuning or zero-shot prompts.

**Prompt Ensemble.** In our work, we use a prompt ensemble together with a noise mechanism for privatization. Previously, ensembling multiple prompts has been explored for improving inference quality with large language models. Pitis et al. (2023) proposed to boost the in-context learning by ensembling multiple few-shot prompts, which is effective on classification tasks. Similarly, Hou et al. (2023) query an LLM by samples enveloped in diverse prompt templates, exhibiting state-of-the-art classification performance. Yet the line of works focuses on few-token prediction instead of long-context generation. As a pioneering effort, the ensemble technique was harnessed for sequence-to-sequence linguistic generation, leveraging long short-term memory networks equipped with attention mechanisms (Juraska et al., 2018). Subsequently, this ensemble paradigm found its application in large language transformers. Hewitt et al. (2021) orchestrated an ensemble of token logits sourced from a lightweight-finetuned language model and a fully-fintuned one to generate texts, thereby bolstering the robustness of the generated content. Different from (Hewitt et al., 2021), our method leans on a voting-based ensemble of prompts from multiple prompts, which facilitates the application of differential privacy. Novel to this work, we show that the ensemble of hundreds of prompts can be effective in generating a long context with DP noise.

## 3 PRELIMINARIES

**Large Language Models (LLMs) and Prompt Tuning.** LLMs like GPT(Radford et al., 2018; OpenAI, 2023), Llama (Touvron et al., 2023), and OPT (Zhang et al., 2022) are pre-trained to generate

tokens conditioned on previous context. Generally, the language generation can be represented as a conditional probability $p_{\text{LM}}^t(y|x)$ where $x$ is a prompt and $y$ is the corresponding output. The temperature $t \geq 0$ can be increased to generate more diverse responses. We use $\pi$ to represent a front-end prompt, e.g., a task instruction that guides the LLM to think and conclude a response. *Prompt tuning* optimizes a prompt $\pi$ that can be wrapped with the input query $x$ in a template $F(\pi, x)$ and improves the response quality of LLM $p_{\text{LM}}^t(y|F(\pi, x))$, e.g., the accuracy on text classification.

**Differential Privacy** (Dwork et al., 2006) stands as the gold standard for assessing the privacy guarantee of machine learning algorithms. DP has gained significant attention among the privacy community as a robust, quantifiable privacy notion, thereby becoming the de-facto choice in privacy protection. Formally, we use $D, D' \in \mathbb{N}^{\mathcal{X}}$ to denote two datasets with an unspecified size over space $\mathcal{X}$. We call two datasets $D$ and $D'$ *adjacent* (denoted as $D \sim D'$) if we can construct one by adding/removing one data point from the other, e.g., $D = D' \cup \{z\}$ for some $z \in \mathcal{X}$.

**Definition 3.1** (Differential Privacy (Dwork et al., 2006)). For $\epsilon, \delta \geq 0$, a mechanism $\mathcal{M} : \mathbb{N}^{\mathcal{X}} \to \mathcal{Y}$ is $(\epsilon, \delta)$-*differentially private* if for every pair of adjacent datasets $D \sim D'$ and for every subset of possible outputs $E \subseteq \mathcal{Y}$, we have $\Pr_{\mathcal{M}}[\mathcal{M}(D) \in E] \leq e^{\epsilon} \Pr_{\mathcal{M}}[\mathcal{M}(D') \in E] + \delta$ here the randomness is over the coin flips of $\mathcal{M}$.

The above definition indicates that for an arbitrary pair of neighboring datasets, a DP algorithm should yield statistically indistinguishable output distribution, preventing adversaries from distinguishing between the outcomes from the datasets. In our study, the mechanism $\mathcal{M}$ being considered is the prompt generation algorithm.

## 4 METHOD

**Assumptions.** Due to the convenience and high performance of cloud models, it is a common interest for a client to tune a prompt that can be served on the cloud. We assume that a client has a set of data $D$ that will be used for prompt tuning but has strict constraints on the data usage as follows. *1) Data Confidentiality.* The client data cannot be shared with the cloud-model vendor. *2) Information Privacy.* The tuned prompt should not leak private information about the client data, including but not limited to enclosing private contents, and inferrable private information. *3) Model Ownership.* On the cloud, model ownership could be a concern and therefore parameters should not be shared with the client.

**Threat Model.** We assume an adversary on the cloud-model vendor side which aims to gain private information (e.g., membership information) from the private dataset stored in the client device. The adversary can only get a tuned prompt provided by the client but can leverage any available LLMs for attacking. The real-world consequence of privacy leakage through released prompts could result in violation of privacy regulation, e.g., GDPR (2016). Concretely, private identifiable information (e.g., names) could be exposed in prompts. Empirically, the privacy risks have been identified in existing works using viable attacks (Wang et al., 2023a; Duan et al., 2023b). Especially, Liu (2023) shows that private instructions behind Bing can be extracted merely by adversarial prompts.

**Main Idea.** To preserve the data confidentiality and privacy, we propose Differentially-Private Offsite Prompt Tuning (DP-OPT) which isolates the prompt tuning and data from the cloud model. The general idea of DP-OPT includes two steps: *1) Private Prompt Engineering*: Engineer a private prompt $\pi$ by fully localized model and datasets, i.e., $\pi \sim \text{DP-OPT}(D, p_{\text{LM}}^t(\cdot))$; *2) Prompt Transfer*: Deploy prompts on cloud model for public inference, i.e., $y \leftarrow p_{\text{cloud-LM}}^t(y|F(x, \pi))$, where $F()$ is a forward template.

To achieve the goal, the two major technical challenges are: (1) How to engineer a model-transferable prompt? (2) How to guarantee that the prompts do not leak private information? We will answer the two questions sequentially in the following two sections.

### 4.1 TRANSFERABLE DISCRETE PROMPTS ENABLE OFFSITE PROMPT TUNING

To make a prompt transferable across models, it is necessary to have a discrete prompt that is not bonded with any model-specific embeddings or tokenization strategies. Importantly, recent advances show that discrete prompts are naturally transferable (to some extent) across domains. Wen et al. (2023) demonstrated that the projected soft prompts tuned by their method, *PEZ*, on GPT-2 (755M) can be used on larger GPT-2 variants (1.3B) or different architectures, like OPT (6.7B). However, such a transfer was shown to suffer from a significant loss of performance. As reported in their paper,

the prompt tuned on GPT-2 (755M) would lose $10.9\%$ absolute accuracy on SST-2 if transferred to OPT (2.7B) and $15.7\%$ to OPT (6.7B). We extend the same experiment to training on Vicuna-7b and testing on DaVinci-003. Similarly, we observe a $6.9\%$ loss of accuracy upon transfer. The situation could worsen on smaller datasets according to our extended experiments in Appendix B.2.

As discussed in (Wen et al., 2023), the key reason for the poor transferability of projected prompt tuning is the incoherence of the tuned prompts. For example, the method for SST-2 with the fluency constraint produced "*negative vibeThis immatureollywood MandarinollywoodThis energetic screenplay.*". This means that the method does not generate a semantically transferable but might still heavily hinge on the embedding space to prompt the training model.

Observing the limitation of projected prompt tuning, we intend to find a semantically transferable prompt. To avoid the pitfalls of the embedding space, we look for a method that does not backward the signal through the embeddings but produces a fluent and coherent prompt. Inspired by the automatic prompt engineering (APE) (Zhou et al., 2022; Sordoni et al., 2023), we conjecture that the LLM itself is an ideal tool for this purpose. Given a well-trained LLM, APE uses samples as context and prompts the LLM to generate a task instruction, which is naturally fluent, coherent, and perhaps transferable. As the task instruction is not optimized in the embedding space but in the output space, it barely relies on hidden neural connections for enhancing inference accuracy. Instead, it captures the explicit and generalizable task semantics that may be reused and strengthened by stronger LLMs. Therefore, we hypothesize that *discrete prompts crafted by one LLM may transfer to another with target-model-dependent performance on the same task*.

**Make LLM Prompt Engineer.** To gain the best performance, we consider the state-of-the-art APE method, Deep Language Network (DLN) (Sordoni et al., 2023), that mimics gradient-based optimization to use forward and backward to train prompts on a dataset $D = \{(x, y)\}$ with input-output pairs $(x, y)$. *1) Prompt Generation.* In the forward pass, an LLM is prompted via a forward template $F(x, \pi)$ to predict labels on a small batch of training samples $S \leftarrow \{(x, y) \sim D\}$,

| Task | Source | Target | | | |
|---|---|---|---|---|---|
| | Vicuna-7b | Llama-2-70b | $\Delta$ | DaVinci-003 | $\Delta$ |
| SST-2 | 92.8(0.2) | 93.3(1.8) | **0.5** | 92.7(0.3) | $-0.1$ |
| Trec | 59.9(5.7) | 65.2(7.5) | **5.3** | 70.7(3.9) | **11.2** |
| Mpqa | 75.8(6.2) | 78.0(2.3) | **2.2** | 81.4(1.6) | **5.6** |
| Disaster | 61.7(3.2) | 73.1(1.6) | **11.4** | 77.0(1.9) | **15.3** |

Table 1: DLN-1 can produce transferable prompts, bringing non-trivial gains ($\Delta$). Accuracy ($\%$) on test sets is reported with standard deviation in the brackets.

i.e., $\hat{y} \sim p^t_{\mathrm{LM}}(y|F(x, \pi))$. Then in the backward pass, the correct and incorrect predictions will be used as in-context examples for LLM to generate a task instruction $\pi$. Formally, $\pi$ is sampled from $p^t_{\mathrm{LM}}(\pi|B_\pi(\{(x, y, \hat{y})\}, \pi))$ where $B_\pi$ is a backward template. *2) Prompt Selection.* With a set of candidate prompts, DLN-1 yields the best prompt with the highest log probability on the training set.

**LLM-Engineered Prompts Are Transferrable.** To verify our hypothesis that the prompts generated by DLN-1 are transferable across models and gain better accuracy, we let DLN-1 train prompts using a relatively small LLM, Vicuna-7b (Chiang et al., 2023). Then, the generated prompt is then applied to a larger homogeneous-architecture model, Llama-2-70b, and a heterogeneous-architecture and closed-source model, DaVinc-003. Experiments are carried out on four sentiment classification tasks. In Table 1, DLN-1 demonstrates a competitive performance on the target model. Different from traditional observation, DLN-1 even attains $8\%$ accuracy gains after transfer to DaVinci-003 on average and $4.9\%$ to Llama-2-70b. Not surprisingly, the transferrable prompts generated by DLN-1 are also coherent and fluent as exemplified in Fig. 2.

## 4.2 Differentially-Private Offsite Prompt Tuning (DP-OPT)

Though leveraging DLN-1 can keep data confidential against the cloud model by transferring prompts, it does not provide any provable guarantee for privacy protection. Notice that DLN-1 feeds private samples to LLM to generate instruction prompts, which may leak private information. To unveil the risk, we present an example prompt from DLN-1 in Fig. 2, where three private examples are verbatim copies from the training set. The only difference between the generated examples and private examples is the capitalization, as LLM tends to make the sentence grammatically correct. Since LLMs are known to repeat words from prompts, it is not surprising that LLMs copy private in-content examples into generated prompts.

> **DLN-1 SST-2 prompt**
>
> Classify the input text as positive or negative. Use the correct output for each input. Avoid phrases like "might" or "probably", "carnage and", "i recommend" or words like "barely". Input: "(1) Actor Michel Serrault" - Correct Output: positive Input: "(2) Unique residences" - Correct Output: positive Input: "(3) Buy the movie milk when the TV cow is free" - Correct Output: negative Input: A
>
> **Leaked training samples**: (1) ▶[actor michel serrault ] (2) ▶[unique residences ] (3) ▶[buy the movie milk when the tv cow is free]

Figure 2: A DLN-1-generated prompt is coherent but suffers from privacy leakage. We highlight the potential leakage in the prompt and semantically-nearest ▶[leaked sample] from the training set.

---

**Algorithm 1** DP-OPT ($\epsilon_0 < \infty$) or OPT ($\epsilon_0 = \infty$)

---

**Input**: Training datasets $D = \{(x, y)\}$ and $D_{\text{val}} = \{(x, y)\}$, LLM $p_{\text{LM}}^t(\cdot)$ with generation temperature $t$, number of prompts $N$, and privacy parameters $\epsilon_0, \delta_0$.

1: Initialize $\pi_0$ with a task description or empty and $\Pi \leftarrow \emptyset$
2: $D' \leftarrow \{(x, y, \hat{y}) | \hat{y} = p_{\text{LM}}^0(y | F(x, \pi)), \forall (x, y) \in D\}$              ▷ Forward pass
3: **for** $n \in \{1, \ldots, N\}$ **do**
4:      $\pi^n \sim \text{DP-EnsGen}(\pi, D', \epsilon_0, \delta_0)$             ▷ Private Prompt Generation
5:      $\Pi \leftarrow \Pi \cup \{\pi^n\}$
6: $\hat{\pi} \leftarrow \text{DP-Argmax}_{\pi \in \Pi}^{\epsilon_0} \text{Accuracy}(\pi; D_{\text{val}}) \cdot |D_{\text{val}}|$        ▷ Private Prompt Selection
7: **Output** $\hat{\pi}$

---

To defend the risk, we develop a DP variant of DLN-1, termed DP-OPT, which provides provable privacy protection through the DP noise mechanism. In DP-OPT, we privatize the two core operations in DLN-1: prompt generation and prompt selection, that directly access private data.

**Private Prompt Generation.** As demonstrated above, the main privacy leakage comes from non-private prompt proposals. In Algorithm 2, we develop a privatized version of the prompt generation. Specifically, we leverage the classic *sample-and-aggregate* paradigm (Nissim et al., 2007), where we partition the full batch of data into disjoint subsets. We then generate each token based on the voting results formed by querying the language model with each disjoint subset. While we can simply apply the commonly used Exponential Mechanism (EM) (McSherry & Talwar, 2007) to privately release the token with the maximum count, the naive application of EM may result in high variance and poor performance as the token space can be as large as 30,000 Chiang et al. (2023). Fortunately, extending EM on large domain space has been studied in the DP community. In this work, we leverage the LimitedDomain mechanism (Durfee & Rogers, 2019) which reduces the domain space to only those tokens with top-$\bar{k}$ vote counts (with some privacy budget). We note that LimitedDomain has a small failure probability that will not output any token for the scenario where the highest vote count is not too high compared with the $\bar{k}$th highest vote count. In this case, we retry to generate using the next batch of data. If we run into more than one failure case for generating a single token, it means that the disjoint partitions do not have a majority agreement on a single token choice and we terminate the token generation for this prompt.

**Private Selection among Generated Prompts.** With the generated prompt candidates, DLN-1 selects the best one by contradicting their performance on training samples. This may leak private information about the validation set when some private samples significantly affect the evaluation. To defend against such risks, we use the exponential mechanism to select the best-generated prompt that achieves the highest count of correct predictions on the validation set in a differentially private manner. Formally, given a histogram $h$, we define DP-Argmax$^\epsilon$ as $\Pr[\text{DP-Argmax}^\epsilon(h) = j] \propto \exp(\epsilon h_j)$. Note that this part protects the privacy of the validation set, which is disjoint with the training set. Hence, the privacy cost of this part does not add up to the privacy cost of prompt generation.

We adopt the commonly used Rényi differential privacy (RDP) (Mironov, 2017) to track the privacy cost of Algorithm 1 for generating prompts and ensure that the total privacy cost is within a prespecified budget. To understand the asymptotic behavior of the number of generated tokens $m$, we provide Theorem A.3 in the appendix showing that the growth of $\epsilon$ is of the order of $\sqrt{m}\epsilon_0$. We defer the detailed privacy analysis to Appendix A.2.

---

**Algorithm 2 DP-EnsGen**: Differentially-Private Ensemble Generation

---

**Input**: Max number of new tokens $L$, privacy parameters: $\epsilon_0, \delta_0$, subsampling rate $q$, and predicted dataset $D'$.

1: Initialize output $z \leftarrow [\ ]$ and $l \leftarrow 0$
2: **for** $l < L$ **do**
3:     **if** $\epsilon < \infty$ or $l = 0$ **then**
4:         Sample a batch $S \leftarrow \{(x, y, \hat{y}) \sim D'\}$ by Poisson sampling with probability $q$
5:         Partition $S'$ into disjoint subsets of equal size ($\cup_{j=1}^{J} S_j = S'$)
6:     $h = \text{Histogram}(\{\tau_j \leftarrow p_{\text{LM}}^t(\pi | B_\pi(S_j, \pi, z)) \text{ for } j \in [J]\})$    ▷ Histogram of the next token
7:     **if** $\epsilon < \infty$ **then**
8:         $\tau_l \leftarrow \text{LimitedDomain}(h; \bar{k} = 10, \epsilon_0, \delta_0)$ (Algorithm 4)     ▷ Select private top-1 token
9:     **else**
10:         $\tau_l = \arg\max_i h_i$
11:     **if** $\tau$ is EOS **then** Break                                  ▷ Append generation
12:     **if** $\tau_l = \perp$ **then**
13:         **if** $l > 0$ and $\tau_{l-1} = \perp$ **then** Break
14:     **else**
15:         $l \leftarrow l + 1$
16: **Output** $[z_0, \cdots, z_l]$

---

Table 2: Test accuracy (%) with standard deviation in the brackets. All trainable methods are trained on Vicuna-7b. **Bold methods** are model-transferable and therefore are tested on DaVinci-003. PromptSGD and PromptDPSGD are not transferable and, thereby are tested on Vicuna-7b. The non-confidential baseline uses private data in prompts. Confidential and confidential-and-private prompts are trained on local Vicuna-7b. We highlight the **best** and the second-best results in each setting as bold and underlined numbers, respectively.

| Method | SST-2 | Trec | Mpqa | Disaster | Average |
|---|---|---|---|---|---|
| **ICL** | 94.7(0.4) | 79.1(0.5) | 88.8(0.1) | 69.0(5.9) | 82.9 |
| PromptSGD (Vicuna-7b) | **93.7(1.1)** | 56.1(6.7) | **88.1(0.8)** | **79.4(1.2)** | 79.3 |
| **DLN-1** | 92.7(0.3) | 70.7(3.9) | 81.4(1.6) | 77.0(1.9) | 80.4 |
| **OPT** (ours) | 92.4(0.5) | **71.5(0.8)** | 85.8(1.2) | 79.0(0.9) | **82.2** |
| PromptDPSGD (Vicuna-7b) | 90.4(1.7) | 32.3(3.1) | 84.2(4.0) | 78.5(0.4) | 70.5 |
| **0-shot** | **92.4(0.0)** | 51.8(0.2) | 84.5(0.1) | 76.4(0.2) | 76.3 |
| **DP-OPT** (ours) | 92.2(0.8) | **68.7(6.5)** | **85.8(0.7)** | **78.9(0.3)** | **81.4** |

## 5 EXPERIMENTS

**Tasks.** Our study focuses on sentiment classification tasks. We use SST-2 from the GLUE benchmark (Wang et al., 2018) which includes $6.7 \times 10^4$ samples. Trec and Mpqa (Lu et al., 2021) and Disaster (Bansal et al., 2019) are smaller datasets consisting of fewer training samples. Both SST2 and Mpqa are sentiment classification tasks for positive or negative reviews. Trec is to predict a 6-option label for question types. Disaster analyzes if the sentence is relevant to disaster.

**Setup.** We use Vicuna-7b as the local model to train prompts by default. For DP algorithms, we follow the common practice to set the privacy budget as $\epsilon = 8$ and $\delta = 1/|D|$ where $|D|$ is the training size (De et al., 2022; Sander et al., 2023; Duan et al., 2023a). More experiment details are deferred to Appendix B.1.

### 5.1 PRIVATE OFFSITE PROMPT TUNING

In Table 2, we evaluate the effectiveness of DP-OPT in generating private prompts for DaVinci-003. Our private baseline is the *PromptDPSGD* which uses DPSGD (Abadi et al., 2016) to tune soft prompts (Duan et al., 2023a). We also include the non-private variant of *PromptDPSGD*, *PromptSGD*, for comparison. As a non-private baseline, we follow (Sordoni et al., 2023) to include the In-Context Learning (*ICL*) with 5 class-balanced demonstrations that have secondary best performance compared to DLN-1 in the sentiment classification. To show the improvement of training, we evaluate the

Table 3: Transfer test accuracy (%) on different models with standard deviation in brackets. Trainable methods (bold) are executed on Vicuna-7b. ICL is represented as an upper bound without confidentiality. We highlight the **best** and the second-best *confidential* methods as bold and underlined numbers, respectively.

| Task | Method | Vicuna-7b | Vicuna-33b | Llama-2-13b | Llama-2-70b | DaVinci-003 | Average |
|---|---|---|---|---|---|---|---|
| SST-2 | ICL | 90.1(1.5) | 94.2(0.1) | 94.9(0.4) | 95.5(0.6) | 94.7(0.4) | 93.9 |
| | **DLN-1** | **92.8(0.2)** | **92.5(2.2)** | **93.5(1.0)** | 93.3(1.8) | **92.7(0.3)** | **93.1** |
| | **OPT** | 89.7(2.7) | 91.8(1.7) | 92.1(0.9) | **94.2(1.3)** | 92.4(0.5) | 92.0 |
| | **DP-OPT** | 89.5(2.6) | 92.5(0.3) | 92.7(1.5) | 93.0(1.6) | 92.2(0.8) | 92.0 |
| Trec | ICL | 49.8(18.9) | 66.9(3.9) | 77.8(7.8) | 72.7(8.7) | 79.1(5.1) | 69.2 |
| | **DLN-1** | 59.9(5.7) | 55.2(8.2) | 38.6(0.5) | **65.2(7.5)** | **70.7(3.9)** | 57.9 |
| | **OPT** | 62.1(1.2) | **72.5(10.2)** | 53.7(14.1) | 52.3(1.6) | 70.4(2.7) | 62.2 |
| | **DP-OPT** | **65.3(4.3)** | 69.7(15.1) | **61.9(2.2)** | 53.6(3.8) | 68.7(6.5) | **63.8** |
| Mpqa | ICL | 85.4(2.0) | 85.6(0.9) | 86.0(0.9) | 87.8(0.2) | 88.8(0.1) | 86.7 |
| | **DLN-1** | 75.8(6.2) | 64.8(8.7) | 66.3(10.3) | 78.0(2.3) | 81.4(1.6) | 73.2 |
| | **OPT** | **80.8(1.2)** | **75.6(1.6)** | 69.5(9.8) | **84.5(1.1)** | **85.8(1.2)** | 79.2 |
| | **DP-OPT** | 80.7(3.3) | 67.8(6.7) | **82.8(1.6)** | 81.7(3.1) | **85.8(0.7)** | **79.8** |
| Disaster | ICL | 58.9(1.3) | 52.0(4.2) | 61.2(3.4) | 69.6(4.5) | 69.0(5.9) | 62.1 |
| | **DLN-1** | 61.7(3.2) | **62.4(4.3)** | 66.3(8.3) | **73.1(1.6)** | 77.0(1.9) | **68.1** |
| | **OPT** | **67.4(5.1)** | 60.6(9.1) | 57.8(10.0) | 49.1(12.0) | **79.0(0.9)** | 62.8 |
| | **DP-OPT** | 65.6(0.3) | 53.7(0.0) | **67.9(1.5)** | 42.9(0.3) | 78.9(0.3) | 61.8 |

initial instruction (*0-shot*) wrapped in the forward template. DLN-1 serves as the state-of-the-art LLM-driven tuning method for offsite transfer.

We demonstrate that offsite prompt tuning via OPT and DP-OPT can significantly enhance prompt efficacy compared to the initial instruction (0-shot). For three tasks (SST-2, Mpqa, and Disaster), OPT and DP-OPT approach the performance of the non-private baseline, ICL. In the absence of DP, OPT boosts performance for these three tasks relative to DLN-1, likely due to the ensemble's ability to bolster model generalization.

While automatic discrete prompts often fall short compared to soft prompts in the literature (Wen et al., 2023), our research indicates that using transfer prompts with DaVinci-003 can somewhat alleviate this discrepancy. When operating within the same DP budget, both OPT and DP-OPT demonstrate superior results compared to PromptDPSGD and are nearly on par with the non-private PromptSGD. On the SST-2 dataset, DP-OPT matches the accuracy of PromptSGD. For Trec, discrete prompts consistently surpass the performance of soft prompts. The underwhelming results of PromptDPSGD can be traced back to weak zero-shot performance on Trec and the noisy gradient descent. However, when the zero-shot approach excels, as seen in our other three datasets, PromptDPSGD will yield better accuracy.

In Table 3, we assess the transferability of the prompts produced by Vicuan-7b on various larger models including Vicuna-33b, Llama-2-13b, Llama-2-70b (Touvron et al., 2023) and DaVinci-003 (text generation version of GPT3.5) (Ouyang et al., 2022). The experiment yields several intriguing implications. 1) The closed-source model, DaVinci-003, exhibits greater stability in transfer compared to its open-sourced counterparts, where DP-OPT presents competitive performance compared to non-private baselines. Such stability offers more reliable predictions in various applications and therefore encourages clients to pair DP-OPT with the closed-source DaVinci-003. 2) Without the DP noise mechanism, the ensemble method (OPT) itself enhances prompt quality relative to DLN-1 on Vicuna-33b and Llama-2-13b. 2) We observe a discrepancy in DLN-1's performance on Trec, which is considerably lower than the figures presented in (Sordoni et al., 2023). It seems that Vicuna-7b struggles with the complexities of the 5-way classification task present in the Trec dataset when engineering prompts. This limitation could be a result of architectural constraints or training nuances specific to Vicuna-7b.

## 5.2 ABLATION STUDIES

**Privacy-utility Trade-off.** In order to examine the privacy-utility trade-off of DP-OPT, we conduct an ablation study with varying privacy parameters $\epsilon$ and report the test accuracy of DP-

---

**OPT SST2 prompt**

Classify the input text as positive or negative. # Student successes Input: "(1) the movie was a masterpiece"
Correct Output: positive Input: "(2) the movie was a disaster" Correct Output

**Leaked training samples**: (1) ▶*[the movie is hardly a masterpiece]* (2) ▶*[the movie is amateurish]* ▶*[the movie is a disaster]*

---

**DP-OPT SST2 prompt with budget $\epsilon = 8$**

Classify the input text as positive or negative. # Student successes Input: "(1) a movie that is a masterpiece"
Correct Output: positive Input: "(2) the movie was a disaster" Correct Output: negative Input

**Leaked training samples**: (1) ▶*[the movie is hardly a masterpiece]* (2) ▶*[the movie is amateurish]* ▶*[the movie is a disaster]*

---

**DP-OPT SST2 prompt with budget $\epsilon = 4$**

Classify the input text as positive or negative. * For positive text: + "(1) The film was a masterpiece." + "(2) a collectively stellar performance" + "(3) The book was a delight"

**Leaked training samples**: (1) ▶*[the movie is hardly a masterpiece]* (2) ▶*[the comedy was funny]* (3) ▶*[is a heartfelt story]* ▶*[was a guilty pleasure]*

---

**DP-OPT SST2 prompt with budget $\epsilon = 2$**

Classify the input text as positive or negative.

---

Figure 3: Examples of Generated Prompts. OPT/DP-OPT tends to generate pseudo examples (blue text) which do not belong to the training set. We highlight potentially-leaked samples and semantically-nearest retrieved ▶*[training samples]* (there might be multiple such samples). More examples are in Table 10.

OPT on the SST-2 dataset with different test models, shown in Fig. 4. In the smallest model, Vicuna-7b, we observe a trade-off between accuracy and privacy: when the privacy budget ($\epsilon$) reduces to 1, the accuracy drops to $86\%$ approximately. The accuracy drops because when the budget is limited, the LimitedDomain will prohibit DP-OPT from generating any new content.

Interestingly, such a trade-off is greatly mitigated when scaling up model sizes. We notice that the LLM can still repeat the initial instructions even if under a very limited budget. Thus, the mitigation can be attained when larger LLMs present a strong zero-shot ability.

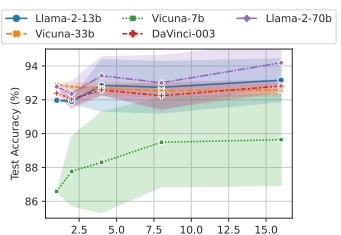

Figure 4: Privacy-utility trade-off of DP-OPT. Smaller $\epsilon$ indicates stricter privacy protection.

**Examples of Privacy Leakage in Generated Prompts.** In Fig. 2 and Fig. 3, we present examples of generated prompts that potentially leak private samples from the training dataset. To find such examples, we perform a semantic similarity search to retrieve the training sentence closest to each forged demonstration (details in Appendix B.1). In the prompts generated by DLN-1 (Fig. 2), we observe verbatim copies of training examples. In contrast, OPT and DP-OPT will modify a few words when generation, potentially due to the randomness of the ensemble. In Fig. 3, The word "*is*" in training example "*the movie is a disaster*" is replaced with "was" in an OPT-generated prompt. Actually, "the movie is" is a common format in the training set, for example, "the movie is amateurish". On the other hand, DP-OPT-generated prompts exhibit privacy-preserving behaviors due to their formal privacy guarantee. Especially, when $\epsilon$ get smaller, there are fewer word copies. Our observations of the generated prompts imply that verbatim copies of training examples are diminished by reducing $\epsilon$ in DP-OPT.

## 6 DISCUSSION AND CONCLUSION

With the rising popularity of prompt tuning, our research endeavors to extend this tool to applications with heightened privacy concerns. We introduce the pioneering end-to-end system designed to derive differentially-private prompts from confidential training datasets and deploy these prompts on cloud models. Our approach is underpinned by theoretical validations of its privacy assurances, and through empirical analysis, we highlight the advantageous balance it strikes between utility and data privacy caused by the strong performance of scaled LLMs.

ACKNOWLEDGMENTS

The work of Z. Wang is in part supported by the National Science Foundation under Grant IIS-2212176. This work is partially supported by the National Science Foundation under grant No. 1910100, No. 2046726, No. 2229876, DARPA GARD, the National Aeronautics and Space Administration (NASA) under grant No. 80NSSC20M0229, the Alfred P. Sloan Fellowship, the Michael Hammer Fellowship at MIT, and Princeton's Gordon Y. S. Wu Fellowship. Portions of this research were conducted with the advanced computing resources provided by Texas A&M High Performance Research Computing[1], a composable computing cluster (He et al., 2023). We also want to thank anonymous reviewers for their constructive suggestions and comments.

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

# A    METHOD

## A.1    DEEP LANGUAGE NETWORK

In Algorithm 3, we present the DLN-1 algorithm from (Sordoni et al., 2023) where we highlight the potential privacy leakage.

---

**Algorithm 3** Deep Language Network (DLN-1) (Sordoni et al., 2023) and  potential privacy leakage .

---

**Input**: Client datasets $D = \{(x, y)\}$, number of prompts $N$, number of iterations $T$

1: Initialize $\pi$ with a task description or empty
2: **for** $t \in \{1, \dots, T\}$ **do**
3:     $S \leftarrow \{(x, y) \sim D\}$                                                                       ▷ Sample minibatch
4:     $\hat{y} \leftarrow p_{\text{LM}}^0(y|F(x, \pi))$ for all $(x, y)$ in $S$                                        ▷ Foward pass
5:     $\Pi \leftarrow \{\pi^n \sim p_{\text{LM}}^{0.7}(\pi|B_\pi(\{x, y, \hat{y}\}, \pi))\}_{n=1}^N$                    ▷ Sample $N$ candidate prompts
6:     $\pi \leftarrow \arg\max_{\pi \in \Pi} \mathbb{E}_{(x,y) \sim D} \log p_{\text{LM}}(y|F(x, \pi))$              ▷ Select the best prompt
7: **Output** $\pi$

---

## A.2    PRIVACY ANALYSIS

Rényi differential privacy (RDP) (Mironov, 2017) is a variant of the standard $(\epsilon, \delta)$-DP that uses Rényi-divergence as a distance metric between the output distributions of $\mathcal{M}(D)$ and $\mathcal{M}(D')$, which is particularly useful in training differentially private machine learning models.

**Definition A.1** (Rényi Differential Privacy (Mironov, 2017))**.** We say that a mechanism $\mathcal{M}$ is $(\alpha, \epsilon_{\mathcal{M}}(\alpha))$-RDP with order $\alpha \in (1, \infty)$ if for every neighboring dataset $D \sim D'$, we have:

$$D_\alpha\left(\mathcal{M}(D)\|\mathcal{M}(D')\right) := \frac{1}{\alpha - 1} \log \mathbb{E}_{o \sim \mathcal{M}(D')} \left[ \left( \frac{\mu_{\mathcal{M}(D)}(o)}{\mu_{\mathcal{M}(D')}(o)} \right)^\alpha \right] \leq \epsilon_{\mathcal{M}}(\alpha) \tag{1}$$

where $\mu_{\mathcal{M}}(\cdot)$ denotes the density function of $\mathcal{M}$'s distribution.

Next, we introduce a strict relaxation of RDP which allows for a small failure probability $\delta$. It is analog to the $\delta$ term in the standard DP definition.

**Definition A.2** (Approximate RDP (Bun & Steinke, 2016; Zhu & Wang, 2022))**.** We say a randomized algorithm $\mathcal{M}$ is $\delta$-approximately $(\alpha, \epsilon_{\mathcal{M}}(\alpha))$-RDP with order $\alpha \geq 1$, if for all neighboring dataset $D, D'$, there exist events $E$ (depending on $\mathcal{M}(D)$) and $E'$ (depending on $\mathcal{M}(D')$) such that $\Pr[E] \geq 1 - \delta$ and $\Pr[E'] \geq 1 - \delta$, and $\forall \alpha \geq 1$, we have

$$D_\alpha\left(\mathcal{M}(D)|E \parallel \mathcal{M}(D')|E'\right) \leq \epsilon_{\mathcal{M}}(\alpha) \tag{2}$$

In this work, we use RDP and approximate RDP for a tighter measure of the privacy cost, as the composition for both privacy notions is trivial. After we obtain the approximate RDP guarantee for the overall mechanism, we can then convert the privacy guarantee back into the standard DP definition (see Bun & Steinke (2016) and Mironov (2017) for the composition and conversion formula for RDP and approximate RDP). In the following, we state the privacy guarantee of individual building blocks for private prompt generation and selection in terms of (approximate) RDP.

### A.2.1    PRIVACY ANALYSIS FOR PRIVATE PROMPT GENERATION

The exponential mechanism takes a utility function $q : \mathbb{N}^{\mathcal{X}} \times \mathcal{Y} \to \mathbb{R}$ and can be thought of as evaluating how good $q(D, y)$ is for an outcome $y \in \mathcal{Y}$ on dataset $D$. In our context, $y \in \mathcal{Y}$ is a potential token to be released and $q(D, y)$ is the vote count for the token $y$ from the disjoint partitions.

**Definition A.3** (Exponential Mechanism (McSherry & Talwar, 2007))**.** Let $\mathbf{EM}_q : \mathbb{N}^{\mathcal{X}} \to \mathcal{Y}$ be a mechanism where for all outputs $y \in \mathcal{Y}$ we have

$$\Pr[\mathbf{EM}_q(D) = y] \propto \exp\left( \frac{\epsilon}{2\Delta(q)} q(D, y) \right)$$

where $\Delta(q)$ is the sensitivity of the quality score, i.e. for all neighboring inputs $D, D'$ we have $\sup_{y \in \mathcal{Y}} |q(D, y) - q(D', y)| \leq \Delta(q)$

**Theorem A.1** (Bun & Steinke (2016))**.** *The exponential mechanism is $\epsilon$-DP, and $(\alpha, \epsilon_{EM}(\alpha))$-RDP s.t. $\epsilon_{EM}(\alpha) := \frac{\alpha}{2}\epsilon^2$.*

We now state the privacy guarantee for LimitedDomain algorithm.

**Theorem A.2.** *Algorithm 4 satisfy $(\epsilon, \delta)$-DP and $\delta$-approximately $(\alpha, \epsilon_{EM}(\alpha))$-RDP.*

*Proof.* This immediately follows from the proof of Theorem 1 from Durfee & Rogers (2019), as LimitedDomain algorithm is essentially an exponential mechanism with an extra step of reducing the domain size, whose privacy cost is being added to the $\delta$ term. $\square$

---

**Algorithm 4 LimitedDomain**$(h; k, \bar{k}, \epsilon_0, \delta_0)$ from (Durfee & Rogers, 2019), where $\Delta_0$ is the $\ell_0$ sensitivity of the utility function, and $\Delta_\infty$ is the $\ell_\infty$ sensitivity of the utility function.

**Input**: Histogram $h$, top-$k$ (by default, $k = 1$) from the $\bar{k} \in [k, d)$ limited domain, privacy parameters $\epsilon_0, \delta_0$

1: Sort $h$ such that $h_{(1)} \geq h_{(2)} \geq \ldots$
2: $h_\perp \leftarrow h_{\bar{k}+1} + 1 + 2\ln(\min\{\Delta_0, \bar{k}, d - \bar{k}\}/\delta_0)/\epsilon_0$
3: $v_\perp \leftarrow h_\perp + \text{Gumbel}(2\Delta_\infty/\epsilon_0)$
4: **for** $j \leq \bar{k}$ **do**
5:     $v_{(j)} \leftarrow h_{(j)} + \text{Gumbel}(2\Delta_\infty/\epsilon_0)$
6: Sort $\{v_{(j)}\} \cup v_\perp$ and let $v_{i_{(1)}}, \ldots, v_{i_{(j)}}, v_\perp$ be the sorted list up until $v_\perp$
7: **Output** $\{i_{(1)}, \ldots, i_{(j)}, \perp\}$ if $j < k$, otherwise $\{i_{(1)}, \ldots, i_{(k)}\}$

---

**Asymptotic Privacy Analysis.** Below, we provide an asymptotic analysis for the overall privacy bound that may facilitate the understanding of the capability of DP-OPT. Theorem A.3 shows the number of generated tokens is essentially limited by the privacy budget.

**Theorem A.3.** *Suppose we set the privacy parameters of LimitedDomain as $\epsilon_0, \delta_0$, then the total privacy bound of our private prompt generation algorithm for generating $m$ tokens is $(\epsilon, m\delta_0 + \delta')$-DP with any $\delta' > 0$ and*

$$\epsilon = O(\sqrt{m \log(1/\delta')}\epsilon_0)$$

*Proof.* This immediately follows from the advanced composition theorem of differential privacy (Dwork et al., 2010). $\square$

We stress that Theorem A.3 is only used for illustrating the asymptotic growth rate of the $\epsilon$ with the number of tokens being generated. We use the RDP-based privacy accountant to numerically calculate $\epsilon$ in the actual implementation, which leads to a much tighter bound and therefore allows generating more tokens.

### A.2.2 PRIVATE SELECTION AMONG GENERATED PROMPTS

While our private selection algorithm is simply an exponential mechanism, we can actually obtain a tighter privacy bound than what is stated in Theorem A.1. We first state the definition of the monotonic utility function.

**Definition A.4** (Monotonic Utility Function)**.** We say that a utility function $q(\cdot, \cdot)$ is monotonic in the dataset if the addition of a data record can either increase (decrease) or remain the same with any outcome, e.g. $q(D, y) \leq q(D \cup \{x\}, y)$ for any input and outcome $y$.

Clearly, in our context, since the utility function is defined as the count of correct predictions on the validation set, the addition of a validation point will not decrease the utility function, and hence our utility function is monotonic in this case. We now have the following tighter privacy guarantee.

**Theorem A.4** (Durfee & Rogers (2019))**.** *The exponential mechanism is $\epsilon/2$-DP, and $(\alpha, \epsilon_{mEM}(\alpha))$-RDP s.t. $\epsilon_{mEM}(\alpha) := \frac{\alpha}{8}\epsilon^2$ if the utility function $q$ is monotonic.*

## A.3 DISCUSSION OF COMPUTATIONAL EFFICIENCY

Our method is quite efficient and feasible for black-box models. Since we do not require gradients but only the forward pass which mimics zeroth order gradients, our method is much more memory efficient for any gradient-based method, including soft prompt tuning. The main computation bottleneck comes from the ensemble. Here, we provide detailed memory and computation analysis of the ensemble. *1) Computation efficiency.* Our method will do ensemble prediction per token which has a similar complexity as inference. Parallelizing the process could speed up the training. *2) Memory efficiency of training.* As we only do inference, the complexity only depends on the context length. We use $k$ demonstrations in meta-prompts, whose complexity is close to a $k$-shot in-context learning. *3) Memory efficiency of inference.* Compared to ICL, our generated prompts are short resulting in low overhead for memory at inference time.

# B    EXPERIMENTAL SUPPLEMENTARIES

## B.1    EXMPERIMENT DETAILS

We present the detailed statistics of all four tasks in Table 4. Disaster has the smallest volume of training data. But Trec is short on samples per class, making it a harder task.

Table 4: Task statistics. Note that different from (Sordoni et al., 2023), we use the full set rather than a small fixed 250-sample subset of the original dataset for testing. The validation set is selected from the training set per random seed. The ratio of validation with respect to the training set is included in brackets.

| Task | # Train | # Valid | # Test | # Class | Description |
|------|---------|---------|--------|---------|-------------|
| SST-2 | $66,674$ | $673$ (1%) | $1,820$ | 2 | Sentiment analysis on movie reviews |
| Trec | $5,452$ | $272$ (5%) | $500$ | 6 | Question type classification |
| Mpqa | $8,603$ | $430$ (5%) | $2,000$ | 2 | Sentiment analysis |
| Disaster | $4,430$ | $221$ (5%) | $1,000$ | 2 | Determine whether a sentence is relevant to a disaster. |

**Hyperparametes.** For DP algorithms, we limit the DP budget as $\epsilon = 8$. and $\delta = 1/|D|$. Detailed parameters for DP-OPT are given in Table 5. As the total $\delta$ is determined by sample size, we mainly tune the $\epsilon_0$ and $\delta_0$ for each dataset such that enough tokens can be generated in DP-OPT. For DP-OPT and OPT on Mpqa, we set the repetition penalty to be $1.1$ to avoid repeated words. For DPSGD, we adopt `dp-transformers` package to reduce the memory overhead caused by gradient clipping (Wutschitz et al., 2022) and tune the hyper-parameters for each dataset. We follow (Duan et al., 2023a) to use the same templates for soft prompt tuning. For all our experiments, we report average and standard deviation from three repetitions with seeds, $\{1, 2, 3\}$. For ICL, the randomness is on the selection of in-context examples. We notice the most influential factor in ICL is the balance of examples in our experiment. We follow DLN-1 to implement the ICL where we sample 5 balanced examples. Our results are comparable to those reported in (Sordoni et al., 2023) in Trec and Mpqa. ICL results on Disaster are different because we used a much larger test set (using the standard test set of Disaster) while Sordoni *et al.* selected a small 100-sample subset for the test. For all trainable methods, we hold out $5\%$ of training data for validation and report accuracy on the original test set.

Table 5: Hyperparamters for DP-OPT and OPT. For batch size, $5 \times 205$ means 205 5-demo meta-prompts. $\epsilon_0$ and $\delta_0$ are the parameters for LimitedDomain in Algorithm 4.

| Experiment | Max new tokens | Batch size | $\epsilon_0$ | $\delta_0$ | temperature |
|------------|----------------|------------|--------------|------------|-------------|
| SST-2 | 50 | $5 \times 205$ | 1.8 | $5 \times 10^{-7}$ | 1.1 |
| Trec | 50 | $3 \times 102$ | 0.8 | $4 \times 10^{-6}$ | 1.1 |
| Mpqa | 50 | $3 \times 102$ | 1.6 | $5 \times 10^{-6}$ | 1.1 |
| Disaster | 50 | $3 \times 102$ | 0.8 | $4 \times 10^{-6}$ | 1.1 |

In Table 6, we list the initial instructions that we used in OPT, DP-OPT, and DLN-1 following (Sordoni et al., 2023).

Table 6: Initial instructions for tasks.

| Task | Classes | Instruction |
|------|---------|-------------|
| SST-2 | {negative, positive} | Classify the input text as positive or negative. |
| Trec | {description, entity, expression, human, location, number} | Read the following question, then choose whether it is about a description, entity, expression, human, location or number. |
| Mpqa | {negative, positive} | Read the following review, then choose whether it is negative or positive. |
| Disaster | {not relevant, relevant} | Read the following sentence, then choose whether it is relevant to a disaster. |

---

**Classification Forward Template $F$**

```
[PROMPT]

[INPUT]

Output:
```

---

**Backward Template $B_\pi$**

```
A student is completing a task that requires producing a text output from a text input. The student
    receives an instruction that describes how to produce the output given each input. The student
    has made some errors. Your task is to improve the instruction such that the student can fix the
    errors.

This was the instruction.

Instruction: [CURRENT INSTRUCTIOON]

# Student successes
Input: [INPUT]
Correct Output: [OUTPUT]

Input: [INPUT]
Correct Output: [OUTPUT]

# Student errors
Input: [INPUT]
Student Output: [OUTPUT]
Correct Ouput: [TARGET]

Improve the instruction to fix the student errors. [MSG]
Improved Instruction:
```

where [MSG] is randomly chosen from

```
- Clarify the instruction by adding few words or a short sentence. Be concise
- Improve the instruction by providing examples on how to solve the task. Be concise.
- Shorten the instruction by removing superflous words or sentences.
- Rewrite the instruction by providing detailed information to avoid ambiguity. Be concise
```

Figure 5: Templates for DLN-1, OPT and DP-OPT.

In Fig. 5, we list the templates used for DLN-1, OPT and DP-OPT. The templates are slightly different from those in (Sordoni et al., 2023) since we use Vicuna-7b instead of GPT to handle the instructions.

**Finding privacy leakage in prompts.** To find strings leaked from the training set, we perform a semantic similarity search to retrieve the training sentence closest to each forged demonstration. Concretely, we perform mean-pooling of the hidden states of the last layer of the BART-large decoder (Lewis et al., 2019) over the token dimension to obtain sentence embeddings for both training examples and the demonstrations generated by prompt tuning. We use Faiss (Johnson et al., 2019) to retrieve the top-5 training examples closest to the demonstrations that appeared in the generated prompts in terms of the $\ell_2$ distance between their embeddings. We then manually examine the retrieved examples to identify training example leakage.

## B.2 ADDITIONAL EXPERIMENTS

| Task | Source Vicuna-7b | Target Llama-2-70b | Δ | DaVinci-003 | Δ |
|------|------|------|------|------|------|
| SST-22 | 80.4(4.0) | 83.9(1.5) | 3.5 | 73.5 | −6.9 |
| Trec | 46.1(0.8) | 25.1(3.5) | −21 | 46.0(1.4) | −0.1 |
| Mpqa | 74.2(5.7) | 60.8(5.6) | −13.4 | 71.8(7.6) | −2.4 |
| Disaster | 59.9(2.0) | 45.9(4.1) | −14 | 55.4(4.0) | −4.5 |

Table 7: PEZ cannot produce transferable prompts, bringing non-trivial losses (Δ) on multiple tasks. Average accuracy (%) on test sets is reported with standard deviation in the brackets.

**Negative Transfer of Projected Prompt Tuning.** In Table 7, we evaluate the transferability of projected prompts, dubbed PEZ, proposed by Wen et al. (2023). We did not report standard deviation on DaVinci-003 and SST-2 since two prompts are not tokenizable by Davinci-003. On both DaVinci-003 and Llama-2-70b, the PEZ-engineered prompts loses a great portion of accuracy after transfer.

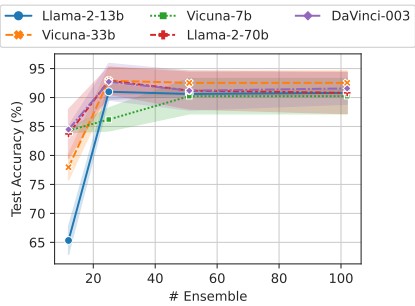

Figure 6: Ablation of the number of ensemble prompts.

**Sensitivity to Ensemble Number.** We ablate the number of ensemble prompts in Fig. 6. For larger models, the number of ensembles seems less influential when the number is larger than 30. Only for Vicuan-7b, the parameter is critical and it is essential to have more prompts in the ensemble. We attribute the stability to the robustness of larger models. Larger models could be less sensitive to words replacement.

**Empirical Evaluation of Privacy Risks.** We evaluate the privacy risks empirically by membership inference attack (MIA) using Likelihood Ratio test (LiRA) (Mireshghallah et al., 2022). Results are reported in Table 8. For SST-2, because of the distributional bias between the training and test sets, we subsample the training set to include samples with more than 20 tokens, in which case only 15 test samples are eliminated. The data filtering can avoid undesired high MIA AUC due to lack of short samples in test sets. We observe non-trivial AUCs for DLN-1 on Mpqa and SST-2. In comparison, both DP-OPT and OPT has very low AUC. OPT has slightly higher risks than DP-OPT when applied on the Trec dataset.

**Comparison to Text Sanitization.** Distinct from our work, text sanitization, for example, Mattern et al. (2022); Utpala et al. (2023), sanitizes text samples before input into LLMs. In Table 9, we compare our method to DLN-1 using sanitized data (Mattern et al., 2022), denoted as *Private DLN-1*.

| Method | Disaster | Mpqa | SST2 | Trec |
|------|------|------|------|------|
| DLN-1 | 0.498 | **0.772** | **0.773** | 0.446 |
| OPT | **0.511** | 0.443 | 0.510 | 0.494 |
| DP-OPT | 0.497 | 0.456 | 0.518 | 0.468 |

Table 8: Average MIA AUC of Likelihood Ratio attacks. Bold numbers indicate the highest leakage risks (over 0.5).

| Data | Private DLN-1 ($\epsilon = 8$) | DP-OPT ($\epsilon = 8$) |
|---|---|---|
| SST-2 | 0.868 | **0.895** |
| Trec | 0.311 | **0.653** |
| Mpqa | 0.690 | **0.807** |
| Disaster | **0.657** | 0.656 |

Table 9: Comparison between DP-OPT and DLN-1 using sanitized data (Mattern et al., 2022). Test accuracy averaged over 3 repetitions are reported on SST-2 using Vicuna-7B.

The implementation includes three steps: **(1)** First, the embedding of each token will be perturbed and projected into the original embedding space. This step can be extended to other sanitization methods (Utpala et al., 2023; Feyisetan et al., 2020). **(2)** We use DLN-1 to tune prompts on these samples. DLN-1 is selected here due to its similarity to our method but can be replaced by other prompt-tuning algorithms in practice. **(3)** We use the generated prompts for inference. Note that the text sanitization is measured under the metric Differential Privacy instead of standard DP. Because of the different privacy assumptions, we emphasize that the meaning of $\epsilon$ is different. We show that our method can outperform Private DLN-1 significantly on three datasets and has similar performance as Private DLN-1 on Disaster.

**Generated Prompts.** In Tables 10 and 11, we give more examples of prompts generated by LLMs. DLN-1 is able to generate a very semantic prompt (e.g., seed 1 in SST-2 task) but may fail to transfer (with a 3.5% drop). Consistent with the conclusions in the main content, the DLN-1 tends to leak more private information and DP-OPT presents much less visible leakage. In the hardest task, Trec, DLN-1 extensively overfits the source model with semantically favored prompts but transfers poorly to DaVinci where two prompts suffer from negative transfer.

Interestingly, we see that "# Student successes Input:" does not provide useful task information but often occurs to enjoy positive transfer, e.g., DP-OPT on SST-2 and Disaster. We conjecture that the prompt induces the LLM to produce "success" output.

We notice that the prompt engineering degrades to generating dummy samples sometimes. However, our method can create prompts without samples and promote the performance, as well. For example, DP-OPT may only slightly change the prompt by appending " # Student successes Input:" to an initial instruction. Intuitively, the modification prompts LLMs to generate "successful" responses. We notice such minor modifications can improve the accuracy of the Disaster dataset from 76.4% (0-shot) to 78.9% (DP-OPT) tested by DaVinci-003. It even outperforms more complicated prompts generated by DLN-1 (77%).

**Avoid Privacy Leakage via Instruction.** When we notice the direct breach of training samples in generated prompts, a straightforward fixture could be to instruct LLMs to keep secrets. We tried two of such instructions in experiments: (1) **Instruction 1**: `Do not provide examples in the prompt.` (2) **Instruction 2**: `Do not use existing samples but create dummy samples as examples in the prompt.` For DLN-1 and (DP-)OPT, we all append the privatization instruction to the instruction in the backward template. We test the instructions on SST-2 following the same setting in the main experiment and report the generated prompts in Table 12. We notice Instruction 1 still leaks private samples and the AUC measured by Likelihood Ratio MIA attack is as high as 73%. Interestingly, the second instruction only generates dummy examples that have no similar examples in the training set. However, the prompt will present non-trivial risks (69% AUC) measured by Likelihood Ratio MIA attack.

In conclusion, though privatization instruction could remove private examples, it still suffers from information leakage. The method is orthogonal to our method that provides theoretical guarantees and can be combined with our DP-OPT to reduce the chance of explicit leakage.

Table 10: Generated prompts. We present test accuracy on Vicuna-7b (Src Val Acc) and DaVinci-003 (Trg Test Acc). Samples found in training sets that best match the generated samples are marked as red text.

| Method | Seed | Src Test Acc | Trg Test Acc | Generated Prompts |
|---|---|---|---|---|
| | | | **SST-2** | |
| **DLN-1** | 1 | 92.7% | 89.2% | 1. Classify the text as positive or negative. 2. The correct output for positive text is positive. 3. The correct output for negative text is negative. 4. Do not classify texts that are a mix of positive and negative as either positive or negative. 5. Pay attention to the nuances of the words used in the text, as some words can be used in both positive and negative contexts. Examples: * Input: The film is a thr |
| | 2 | 92.7% | 93.5% | 1. Film with a small budget Input: * small, personal film with emotional wallop (turn out a small , personal film with an emotional wallop) Output: positive | * Input: nightmare about bad cinema (nightmare about bad cinema ) Output: negative | * Input: film with their charisma (the film with their charisma ) Output: positive | * Input: who feels acting is heart and soul of cinema (who feels acting is the heart and soul of cinema ) Output: positive | * Input: easily one of the best and most exciting movies of the year (easily one of the best and most exciting movies of the year . ) Output: positive | * Input: |
| | 3 | 93.0% | 93.0% | Classify the input text as positive or negative. Use the correct output for each input. Avoid phrases like "might" or "probably", "carnage and", "i recommend" or words like "barely". Input: Actor Michel Serrault (actor michel serrault ) - Correct Output: positive Input: Unique residences (unique residences ) - Correct Output: positive Input: Buy the movie milk when the TV cow is free (buy the movie milk when the tv cow is free) - Correct Output: negative Input: A |
| **OPT** | 1 | 91.7% | 92.0% | Classify the input text as positive or negative. # Student successes Input: "a movie that is a movie that is a movie" (a movie is more than a movie . ) Correct Output: negative Input: "a movie that is a movie that is |
| | 2 | 86.6% | 92.3% | Classify the input text as positive or negative. |
| | 3 | 90.7% | 93.0% | Classify the input text as positive or negative. # Student successes Input: "the movie was a masterpiece" (the movie is hardly a masterpiece ) Correct Output: positive Input: "the movie was a disaster" (the movie is a disaster . ) Correct Output |
| **DP-OPT** | 1 | 90.1% | 91.4% | Classify the input text as positive or negative. # Student successes Input: |
| | 2 | 85.6% | 92.4% | Classify the input text as positive or negative. |
| | 3 | 91.7% | 92.9% | Classify the input text as positive or negative. # Student successes Input: "a movie that is a masterpiece" (a great movie it is not ) Correct Output: positive Input: the movie was a disaster (the movie is a disaster . ) Correct Output: negative Input |
| | | | **Trec** | |
| **DLN-1** | 1 | 57.6% | 58.0% | 1. Read the question carefully. 2. Determine the correct output type based on the prompt. 3. Select the correct output category from the list of options. 4. If the question doesn't have a single correct output, provide a brief description. 5. Avoid ambiguity in the question prompt. |
| | 2 | 55.8% | 51.6% | * Read the question * Choose the correct output from the following options: + Description (D) + Entity (E) + Expression (EX) + Human (H) + Location (L) + Number (N) Correct Outputs: * Human (H) * Human (H) * Human (H) * Number (N) * Human (H) * |
| | 2 | 66.4% | 58.0% | 1. Read the text input carefully. 2. Identify the type of question from the options provided. 3. Choose the correct output type from the list of options. 4. Compare your choice with the correct output provided. 5. If your choice is correct, proceed to the next input. If not, go back to step 2. |
| **OPT** | 1 | 60.4% | 70.6% | * Read the following question, then choose the correct type of output: description, entity, expression, human, location, or number. * For each question, you will be given a text input. Your task is to produce the correct |
| | 2 | 60.4% | 72.0% | * Read the following question, then choose the correct type of output: description, entity, expression, human, location, or number. * For each question, there is only one correct output. * The student must read the question |
| | 3 | 62.8% | 72.0% | * Read the following question, then choose the correct type of output: description, entity, expression, human, location, or number. * For example, if the question is "What is the capital of France?" (What is the capital of Italy ?) (What is the capital of California ?) (What is the largest city in Germany ?), the correct output |
| **DP-OPT** | 1 | 63.0% | 72.2% | * Read the following question, then choose the correct type of output: description, entity, expression, human, location, or number. * For example, if the question is "What is the capital of France?" (What is the capital of Italy ?) (What is the capital of California ?) (What is the largest city in Germany ?), the correct output |
| | 2 | 70.2% | 61.2% | Read the following question, then choose the correct type of output |
| | 3 | 63.0% | 72.4% | * Read the following question, then choose the correct type of output: description, entity, expression, human, location, or number. * For example, if the question is "What is the capital of France?" (What is the capital of Italy ?) (What is the capital of California ?) (What is the largest city in Germany ?), the correct output |

Table 11: Generated prompts. We present test accuracy on Vicuna-7b (Src Val Acc) and DaVinci-003 (Trg Test Acc).

| Method | Seed | Src Test Acc | Trg Test Acc | Generated Prompts |
|--------|------|--------------|--------------|-------------------|
| | | | | Mpqa |
| **DLN-1** | 1 | 82.8% | 79.8% | Read the following review, then choose whether it is negative or positive by identifying the correct output for each input based on the following examples: * Displayed unrelenting resolve and confidence (has displayed unrelenting resolve and confidence): positive * Protests (protests): negative * Constructive and cooperative ties (constructive and cooperative ties): positive * Increasingly angry (increasingly angry): negative * Positive and optimistic views: positive * Denied (denied): negative * Advanced: negative * United States is threatening |
| | 2 | 73.7% | 82.9% | 1. Read the following review. 2. Identify each sentence that requires the student to choose the correct output based on the given input. 3. For each identified sentence, write the correct output based on the given input. 4. Compare your output with the correct output provided in the instructions and make sure they match. 5. If the student's output is incorrect, revise it based on the correct output provided in the instructions. |
| | 2 | 70.9% | 81.6% | 1. Read the following review. 2. Choose whether it is negative or positive. 3. Correct Output: positive 4. Correct Output: positive 5. Correct Output: negative 6. Correct Output: negative 7. Correct Output: negative 8. Correct Output: negative 9. Correct Output: negative 10. Correct Output: negative 11. Correct Output: negative 12. Correct Output: negative 13. Correct Output |
| **OPT** | 1 | 82.3% | 87.3% | Read the following review, then choose whether it is negative or positive. * For each statement, determine if it is negative or positive. |
| | 2 | 80.1% | 85.2% | Read the following review, then choose whether it is negative or positive. * For each statement, determine if it is a positive or negative sentiment. |
| | 3 | 80.1% | 85.1% | Read the following review, then choose whether it is negative or positive. * For each statement, determine if it is a positive or negative sentiment. |
| **DP-OPT** | 1 | 84.6% | 85.0% | Read the following review and determine if it is positive or negative based on the words used in the text. |
| | 2 | 78.8% | 86.3% | Read the following review and determine if it is positive or negative. |
| | 3 | 78.8% | 86.1% | Read the following review and determine if it is positive or negative. |
| | | | | Disaster |
| **DLN-1** | 1 | 58.9% | 76.0% | Read the sentence and determine if the information is relevant to a disaster. The relevant information is when the sentence mentions a disaster or its effects. Please choose "yes" if the sentence discusses a disaster or its effects, and "no" otherwise. |
| | 2 | 65.2% | 75.8% | 1. Read each sentence carefully, and ensure it relates to a disaster or not. Choose "yes" or "no" as the correct output. Produce the correct output for each sentence. |
| | 2 | 60.9% | 79.2% | 1. Read the given sentence. 2. Determine if the sentence is relevant to a disaster. 3. If the sentence is relevant to a disaster, select "yes." If not, select "no." |
| **OPT** | 1 | 66.8% | 79.2% | Read the following sentence, then choose whether it is relevant to a disaster. # Student successes Input: @syeda_khan Wow! I'm so glad I found this! Correct Output: |
| | 2 | 47.0% | 78.1% | Read the following sentence, then choose whether it is relevant to a disaster. # Student successes Input: @Airbnb is a great way to make money. Correct Output: no Input: The world |
| | 3 | 59.6% | 79.8% | Read the following sentence, then choose whether it is relevant to a disaster. # Student successes Input: @#$%&*()_+-=[];':<>| /.? 123 |
| **DP-OPT** | 1 | 65.4% | 78.6% | Read the following sentence, then choose whether it is relevant to a disaster. # Student successes Input: |
| | 2 | 65.4% | 78.9% | Read the following sentence, then choose whether it is relevant to a disaster. # Student successes Input: |
| | 3 | 65.9% | 79.2% | Read the following sentence and determine whether it is relevant to a disaster. # Student successes Input: |

Table 12: Generated prompts. We present test accuracy on Vicuna-7b (Src Val Acc) and DaVinci-003 (Trg Test Acc).

| Method | Seed | Src Test Acc | Trg Test Acc | Generated Prompts |
|---|---|---|---|---|
| | | | | Instruction 1 |
| **DLN-1** | 1 | 91.8% | 91.8% | 1. For each input, produce a corresponding correct output. 2. The input is given first, followed by the correct output. 3. The correct output is a simple and clear statement of the sentiment expressed in the input, for example "positive," "negative," "interesting," etc. 4. Label student errors in red and provide an alternate correct output in blue. 5. Provide examples in parentheses. Examples: # Student successes |
| | 2 | 86.1% | 93.9% | Instruction: Classify the input text as positive or negative. For example: * real-life persona (real-life persona): positive * by a pack of dogs who are smarter than him (by a pack of dogs who are smarter than him): negative * candid, archly funny and deeply authentic (candid , archly funny and deeply authentic): positive * brian tufano 's handsome widescreen photography and paul grabowsky 's excellent music turn this fairly parochial melodrama into something really rather special . (brian tufano 's handsome widescreen photography and paul grabowsky 's excellent music turn this fairly parochial melodrama into something really rather special .): positive |
| | 3 | 91.9% | 93.9% | 1. Classify the input text as positive or negative. 2. Correct Output: (choose one) positive, negative 3. Input: (provide the text input) 4. Correct Output: (choose one) positive, negative. Example: * Input: "The movie is a masterpiece." Output: positive * Input: "This movie is terrible." Output: negative * Input: "I hated the movie |
| | | | | Instruction 2 |
| **DLN-1** | 1 | 90.3% | 93.1% | * Classify each input text as positive or negative based on the given output. Instruction: * Classify the input text as positive or negative. Improved Instruction: * Classify each input text as positive or negative based on the given output. Instruction: * Classify the input text as positive or negative and provide the correct output. Instruction: * Classify each input text as positive |
| | 2 | 92.2% | 94.7% | 1. Classify the text as positive or negative by identifying the sentiment expressed in the text. 2. Identify the main keywords in the text that indicate the sentiment. 3. For each input, provide a clear example of the correct output. For example: * Correct Output: negative 1. "I can't stand this movie. It's so boring and poorly made." (so poorly plotted and scripted . ) * Correct Output: positive 2. "I find the concept of infinity fasc |
| | 3 | 91.0% | 91.8% | Classify the input text as positive or negative. |

