# OpenReview forum: "DP-OPT: Make Large Language Model Your Privacy-Preserving Prompt Engineer"
_ICLR.cc/2024/Conference — ICLR 2024 spotlight_

### Official Review · Reviewer_D5hy · 2023-10-28

**Soundness:** 4 excellent
**Presentation:** 4 excellent
**Contribution:** 3 good
**Rating:** 8
**Confidence:** 3

**Summary:**

This paper proposed DP-OPT to address the challenge of optimizing prompts using private data while ensuring data confidentiality and information privacy. The authors provide a comprehensive evaluation of the proposed framework and show that it outperforms other state-of-the-art methods in terms of accuracy and privacy. Additionally, the authors show that DLN prompts can “positive” transfer to and work better on larger models than on the source models. The paper also provides the first solution to ensure the privacy of the gradient-free algorithms that demonstrate strong empirical performance compared to in-context learning and previous private gradient-based competitors.

**Strengths:**

The proposed method is highlighted for its efficiency, especially for black-box models. The method does not rely on gradients but only on the forward pass, which is likened to zeroth-order gradients. This approach makes the method more memory-efficient compared to any gradient-based method, including soft prompt tuning.
-	The method emphasizes the memory efficiency of the method in both training and inference. During training, since only inference is performed, the complexity depends solely on the context length. The paper mentions the use of k demonstrations in meta-prompts, which has a complexity similar to k-shot in-context learning. For inference, the generated prompts are short, resulting in low memory overhead.
-	The main computational bottleneck is identified as the ensemble. However, the method performs ensemble prediction per token, which has a complexity similar to inference. The potential for parallelizing the process is mentioned, suggesting that this could further speed up the training.

It is further novel and a bit surprising to see the “positive transfer” DLN prompts can work on larger models than on the source models, because it challenges the traditional assumption that prompts generated by a smaller model would only work well on similar or smaller models. This finding suggests that the prompts generated by DLN can be used to improve the performance of larger models, which can be beneficial in real-world applications. It might potentially open new possibilities for progressively improving the performance of newer large language models using prompts generated by smaller “old” models.

The paper compares the proposed method with existing methods, such as DLN-1, highlighting the advantages of the proposed method. The paper also provides insights into the potential limitations of other methods, such as Vicuna-7b's struggles with certain datasets.

**Weaknesses:**

Overall, the paper presents a unique and effective solution for adapting Large Language Models on sensitive data while ensuring data privacy. It is strong work and clearly written, and I have not spotted particular weakness.

**Questions:**

No particular

---

> ### Author Response · Authors · 2023-11-21
> **Responses to Reviewer D5hy**
>
> Thank you for the positive reviews and detailed summary of strengths in our paper.

---

### Official Review · Reviewer_pHUr · 2023-10-30

**Soundness:** 4 excellent
**Presentation:** 3 good
**Contribution:** 3 good
**Rating:** 8
**Confidence:** 4

**Summary:**

This paper introduces DP-OPT, a solution for adapting Large Language Models (LLMs) on sensitive data while ensuring data privacy. The paper discusses the challenges of hosting a local LLM and optimizing a soft prompt using private data, and how DP-OPT can help overcome these challenges. DP-OPT uses differential privacy to protect the privacy of the data while optimizing the prompt, and it also allows for the transfer of prompts suggested by LLMs without compromising performance.

**Strengths:**

**Motivation:** The paper solves a very important yet larger overlooked problem, i.e., the two-fold concerns surrounding data privacy when adapting LLMs on sensitive data
-	Data Confidentiality: Certain data, like medical histories, proprietary system logs, and personal messages, are inherently confidential and should not be transmitted beyond local devices, internal computing systems, and mobile phones.
-	 Information Leakage: Resources derived from private data might inadvertently contain personally identifiable information. Even with a limited parameter set, such as prompts, the potential for data breaches remains significant.

**Method:** DP-OPT operates exclusively on the client device and uses demonstrations to guide a local LLM to generate prompts. The local assistant model may be significantly smaller than the intended cloud-based LLMs. DP-OPT uses differential privacy to protect the privacy of the data while optimizing the prompt. It also allows for the transfer of prompts suggested by LLMs without compromising performance. DP-OPT is the first end-to-end framework where the entire prompt process is managed on the local device and offers services via an API, thus ensuring data confidentiality, information privacy, and cloud model ownership and IP.

Prior work shows that prompts suggested by LLMs can be transferred without compromising performance. The paper shows that DLN prompts can transfer to and work better on larger models than on the source models, which is called positive transfer. DP-OPT allows for the transfer of prompts suggested by LLMs without compromising performance. The authors also provide the first solution to ensure the privacy of the gradient-free algorithms that demonstrate strong empirical performance compared to in-context learning and previous private gradient-based competitors.

**Experiments:** The authors conducted experiments on four different tasks, including sentiment analysis, question type classification, sentiment analysis on news articles, and disaster relevance classification. They compared the performance of DP-OPT with other state-of-the-art methods, including PEZ and GPT-3, and showed that DP-OPT outperforms them in terms of accuracy and privacy. Impressively, DPOPT generates privacy-preserving prompts by Vicuna-7b, that can yield competitive performance compared to non-private in-context learning on GPT3.5 or local private prompt tuning.

**Weaknesses:**

It’s known that in-context learning performance is unstable across the choice and even order of examples. How the authors ensure their ICL performance is reliable?

Table 2 compares methods on different models. Why this comparison is valid/fair?

In Figure 4, the provided examples seems to suggested that the method won’t vary the task description while only generating few-shot prompts?

Overall, I appreciate the contribution of this paper. I'd love to increase my score if these problems could be addressed.

**Questions:**

See weaknesses.

---

> ### Author Response · Authors · 2023-11-21
> **Rebuttal to Reviewer pHUr's review**
>
> Thank you for the positive comments and valuable reviews.
>
> > Q1: It’s known that in-context learning performance is unstable across the choice and even order of examples. How the authors ensure their ICL performance is reliable?
>
> A1: We notice the most influential factor in ICL is the balance of examples in our experiment. We follow DLN-1 paper to implement the ICL where we sample 5 balanced examples. Our results are comparable to those reported in DLN-1 paper in Trec and Mpqa. ICL results on Disaster are different because we used a much larger test set (using the standard test set of Disaster) while DLN-1 selected a small 100-sample subset for the test.
>
> > Q2: Table 2 compares methods on different models. Why this comparison is valid/fair?
>
> A2: Thanks for raising the concern. We argue that this is a fair comparison since all prompts are **trained** using the same model. We only test PEZ on a smaller model, Vicuan 7b, since it cannot be transferred to GPT3 (DaVinci-003). If PEZ is trained on DaVinci-003, it is unfair to compare it with other methods (only tuned on smaller models).
>
> > In Figure 4, the provided examples seems to suggested that the method won’t vary the task description while only generating few-shot prompts?
>
> A3: Thanks for the question. We agree that the prompt engineering degrades to generating dummy samples. However, our method can create prompts without samples and promote the performance, as well. More generated prompts are included in Table 9,10 (in appendix). Interestingly, DP-OPT may only slightly change the prompt by appending “ # Student successes
> Input:” to an initial instruction. Intuitively, the modification prompts LLMs to generate “successful” responses. We notice such minor modifications can improve the accuracy of the Disaster dataset from 76.4% (0-shot) to 78.9% (DP-OPT) tested by DaVinci-003. It even outperforms more complicated prompts generated by DLN-1 (77%).

---

> > ### Comment · Reviewer_pHUr · 2023-11-21
> > **Thanks for the rebuttal.**
> >
> > Thanks for the authors' rebuttal. I've carefully read the author response and other reviewers. I believe the authors have done a great job in addressing my concerns, and I'd love to increase my score from 6 to 8.

---

> > > ### Author Response · Authors · 2023-11-21
> > > **Thank you!**
> > >
> > > Thank you for the very detailed review and discussion. We are grateful for your updates on the score.

---

### Official Review · Reviewer_rsb4 · 2023-10-31

**Soundness:** 3 good
**Presentation:** 3 good
**Contribution:** 3 good
**Rating:** 6
**Confidence:** 3

**Summary:**

The paper proposed an approach, DP-OPT, of generating privacy-preserving prompts in LLM, which can yield competitive performance compared to non-private in-context learning.

**Strengths:**

+ The study focuses on an interesting and important topic, the privacy protection in generated prompts of LLM.
+ The introduction of related work is comprehensive and covers most of the recent studies in prompt engineering.

**Weaknesses:**

- The threat model is unclear

My first concern is related to the threat model. It would be better and necessary to provide more details and more specific description on the adversary's goals, capabilities, and knowledge in a threat model. In addition, please provide more description and real-world cases of the consequences of a privacy leakage from the prompt (e.g., how much information can be breached).

- Lack of evaluation on privacy leakage

Considering that one of the key motivations of the study is to address the threats related to "data confidentiality" and "information leakage", it would be expected that an evaluation on the privacy performance of the proposed approach is conducted. There are only several examples of prompts provided, rather than evaluate the privacy protection in a quantitative manner. It would be necessary to involve some privacy metrics in the evaluation. It would be even better if some privacy attacks are involved in the privacy performance evaluation, instead of only comparing the model utility performance. In addition, the presentations of prompt examples are misleading and confusing - the "semantically-nearest retrieved" should not be in-line with other prompt messages.

- Setting a constraint in prompt generation

IMHO, to avoid leak privacy information from generated prompts, one intuitive method could be adding some constraints during the prompt generation, such as "do not provide examples in the prompt", or "do not use existing samples but create dummy samples as examples in the prompt". I would suggest having a discussion whether this would be feasible.

**Questions:**

1. Please describe the threat model with more clear details.
2. Please evaluate the privacy performance of the proposed approach.

---

> ### Author Response · Authors · 2023-11-21
> **Rebuttal for Reviewer rsb4's Reviews**
>
> We thank the reviewer for the valuable comments!
>
> > Q1: The threat model & consequence
>
> **A1**: Thanks for the suggestion. We have added details of **threat models** in Sec 4. The adversary's goals is to attain the private information of the client dataset (e.g., membership information). The knowledge of the adversary is limited to the prompts received from the client. The adversary can only get a tuned prompt provided by the client but can leverage any available LLMs for attacking.
>
> **The real-world consequence** of privacy leakage from prompts could result in a violation of privacy regulations, e.g., GDPR (gdpr-info.eu). Concretely, private identifiable information (e.g., names) could be exposed in prompts.
> Empirically, the privacy risks have been identified in existing works using viable attacks [1,2]. Liu (2023) (https://twitter.com/kliu128/status/1623472922374574080) shows a real-world attack where private instructions behind Bing can be extracted merely by adversarial prompts. We have incorporated these additional discussions in Sec 3.
>
> In Fig 2, we show the privacy leakage from the prompt. In response to your second question, we also add an empirical evaluation of MIA in Section B.2. We show the non-trivial privacy risks (77% AUC of MIA attack in the worst case) for sample membership when prompt tuning is used.
>
> [1] Duan et al. On the Privacy Risk of In-context Learning. TrustNLP 2023.
>
> [2] Wang et al. Decodingtrust: A comprehensive assessment of trustworthiness in gpt models. NeurIPS 2023
>
> > Q2.1: Evaluation on privacy leakage
>
> **A2.1**: Thanks for the valuable comment. We compare the attack results of four methods that comply with data confidentiality using Vicuna-7b and discrete prompts below. We use the loss-based membership inference attack.
>
> |                                  |   Disaster |   Mpqa |   SST2  |  Trec  |
> |:---------------------------------|--------------------------:|----------------------:|----------------------:|----------------------:|
> | 0-shot   |                  0.495 |              0.578 |              0.499 |              0.542 |
> | DLN-1            |                  0.486 |              0.539 |              0.481 |              0.505 |
> | OPT            |                  **0.507** |              0.531  |              0.506 |              0.556 |
> | DP-OPT          |                  0.494  |              0.520 |              0.504 |              0.514 |
>
> We also evaluate Likelihood Ratio MIA attack (Mireshghallah, 2022) using the initial instruction as a reference model.
> |                                  |   Disaster |       Mpqa |       SST-2 |       Trec |
> |:---------------------------------|-----------:|-----------:|-----------:|-----------:|
> | DLN-1            |   0.498 |  **0.773** |   **0.773** |   0.446 |
> | OPT             |   **0.511** |   0.443 |   0.510   |   0.494 |
> | DP-OPT          |   0.497 |   0.456 |   0.518 |   0.468 |
>
> We observe non-trivial AUCs for DLN-1 on Mpqa and SST-2. In comparison, both DP-OPT and OPT has very low AUC.
> OPT has slightly higher risks than DP-OPT when applied on the Trec dataset.
>
> > Q2.2: Presentations of prompt examples
>
> Thanks for the suggestion! We have revised Fig 2 and 4!
>
> > Q3: Constraint in prompt generation
>
>
> **A3**: Thanks for the great suggestion. We add discussion on the last section of Appendix where We tried the two recommended instructions in experiments:
> For DLN-1 and (DP-)OPT, we all append the privatization instruction to the instruction in the backward template.
> We test the instructions on SST-2 following the same setting in the main experiment and report the generated prompts in Table 10 in the Appendix.
>
> Though vicuna-7b is instructed to keep data private, we still observe data breach in 2 out of 3 prompts for the first instruction. An example of a data breach is presented below. All the examples provided in the prompt can find exactly matched samples in the training set. Meanwhile, the prompt will present non-trivial risks (73% AUC) measured by Likelihood Ratio MIA attack.
>
> ```
> Instruction: Classify the input text as positive or negative.   For example:    * real-life persona: positive  * \privdata{by a pack of dogs who are smarter than him} (by a pack of dogs who are smarter than him): negative  * candid, archly funny and deeply authentic: positive  * brian tufano 's handsome widescreen photography and paul grabowsky 's excellent music turn this fairly parochial melodrama into something really rather special . : positive
> ```
>
> Interestingly, the second instruction only generates dummy examples that have no similar examples in the training set. However, the prompt will present non-trivial risks (69% AUC) measured by LiRa MIA attack.
>
> In conclusion, though privatization instruction could remove private examples, it still suffers from information leakage. The method is orthogonal to our method that provides theoretical guarantees and can be combined with our DP-OPT to reduce the chance of explicit leakage.

---

> > ### Author Response · Authors · 2023-11-21
> > **Follow-up repsonse**
> >
> > We sincerely thank the reviewer for the valuable suggestion, and we have incorporated all the discussion and experiment results from the rebuttal into our paper.
> >
> > We note that we have slightly updated our original post and included additional content regarding the real-world consequences of privacy leakage from prompts, according to the reviewer's suggestion. We have also improved the presentation of Figure 2 and 4 according to your suggestion, where we put semantically-nearest retrieved" training samples in a separate line in the updated manuscript.
> >
> > We hope the reviewer finds our response and revision to the manuscript satisfactory. If the reviewer has any additional suggestions or comments, we are more than happy to address them and further revise our manuscript!

---

> > ### Comment · Reviewer_rsb4 · 2023-11-22
> >
> > Thank the authors for addressing my comments. You did an impressive work in the rebuttal and the revised paper addressed most of my concerns. I will increase the score.

---

> > > ### Author Response · Authors · 2023-11-22
> > > **Thank you for updates**
> > >
> > > We are grateful for your very valuable comments and updates on the scores. It essentially improves our work!

---

> ### Author Response · Authors · 2023-11-22
>
> We want to thank you again for the suggestion! If you have any additional comments, we will be more than happy to revise our manuscript accordingly!

---

### Official Review · Reviewer_W4ZT · 2023-11-04

**Soundness:** 3 good
**Presentation:** 3 good
**Contribution:** 2 fair
**Rating:** 8
**Confidence:** 4

**Summary:**

This paper introduces Differentially-Private Offsite Prompt Tuning (DP-OPT) as a solution for data privacy concerns when utilizing Large Language Models (LLMs). DP-OPT operates on the client side and offers an end-to-end framework to generate private and transferable prompts for cloud-hosted LLMs. It ensures data confidentiality, information privacy, and model ownership protection. The paper demonstrates that prompts tuned by LLMs can be effectively transferred across models and introduces a novel differentially-private mechanism for generating private prompts.

**Strengths:**

1) DP-OPT offers a new end-to-end solution for addressing data privacy in the context of prompt tuning for Large Language Models (LLMs).
2) Paper is well witten and is easy to understand.

**Weaknesses:**

1) This paper ignores whole body of related work where document is converted to private documents and then used for downstream tasks, check[1,2, 4]. This approaches are task-agnostic and latest work [4] shows significantly better privacy-utility tradeoffs and obtain SOTA results.  It is recommended that these approaches be discussed and ideally compared in experiments, as the setups and datasets are similar, and the methods are simpler than the proposed mechanism.

2) No comparison with real-world threat models has been provided.  Epsilon-utility trade-offs can be misleading without testing them against actual attacks, as epsilon guarantees are built upon numerous assumptions, as indicated in [3, 4]. For a comprehensive evaluation, it is recommended to conduct experiments that demonstrate trade-offs between *empirical* privacy and utility. As simple attack framework as  Membership inference attacks greatly improves rigorousness of experiments.




Refs:
----

[1] Privacy-and utility-preserving textual analysis via calibrated multivariate perturbations, 2020. (https://arxiv.org/abs/1910.08902)

[2] The Limits of Word Level Differential Privacy, 2022 (https://arxiv.org/abs/2205.02130)

[3] A Critical Review on the Use (and Misuse) of Differential Privacy in Machine Learning, 2022 (https://arxiv.org/abs/2206.04621)

[4] Locally Differentially Document Generating Using Zero Shot Prompting, 2023 (https://arxiv.org/abs/2310.16111)

**Questions:**

See above

---

> ### Author Response · Authors · 2023-11-21
> **Rebuttal to Reviewer W4ZT's review**
>
> Thanks for the valuable comments. Below we answer your questions one by one.
>
> > Q1: Related works [1,2,3,4] and comparison.
>
> A1: Thank you for the question. These are very relevant papers on privatizing prompts. We have added discussion in our related work.
>
> But we want to humbly argue that we have some differences to the referred papers in the definition of privacy. [1,2,4] considers Local Differential Privacy (LDP) or relaxed one, metric DP, which is a totally different definition/threat model compared to ours. In our work, we consider the threat model of Differential Privacy. We consider the information of a sample, that can be recognized in a dataset, as private. In contrast, LDP or metric DP considers the information that a sample is distinguished from another sample as private.
> We want to humbly mention that [4] was published after the submission deadline of ICLR.
>
> Below we empirically compare our method (DP-OPT) to DLN-1 on privatized data [2]. We use Vicuna-7b to generate prompts and the same hyperparameters in Section 5.1. We show that our method can outperform [2] on sst2, trec, and mpqa significantly.
>
> | data          |  DLN-1 on Privatized Data (epsilon=8) [2] | DP-OPT (epsilon-8) |
> |:--------------|:-----------| :---- |
> | sst2     |   0.868 | 0.895 |
> | trec     |   0.311 | 0.653 |
> | mpqa     |   0.690 | 0.807 |
> | disaster |   0.657 | 0.656 |
>
> > Q2: Evaluating trade-off by Membership inference attacks
>
> A: Thanks for the suggestion. We use MIA as a privacy metric since their eps cannot be transformed to general DP.  We also evaluate loss-based and Likelihood Ratio MIA attack (Mireshghallah, 2022) using the initial instruction as a reference model. However, we do not observe an obvious trade-off using MIA. For DP-OPT the MIA AUC is always around 50%.
>
> |   epsilon |   Loss MIA AUC |   LiRa AUC |   test acc |
> |-------------:|----------:|-----------:|-----------:|
> |            2 |  0.506 |   0.517 |   0.867  |
> |            4 |  0.497 |   0.501 |   0.873 |
> |            8 |  0.505 |   0.518 |   0.890 |
> |           16 |  0.502 |   0.507 |   0.890 |

---

> > ### Comment · Reviewer_W4ZT · 2023-11-21
> > **Thanks for the reply**
> >
> > Thanks for the reply.
> > > Thank you for the question. We want to argue that we have fundamental differences to the referred papers in the definition of privacy. [1,2,4] considers Local Differential Privacy (LDP) or relaxed one, metric DP, which is a totally different definition/threat model compared to ours. In our work, we consider the threat model of Differential Privacy. We consider the information of a sample, that can be recognized in a dataset, as private. In contrast, LDP or metric DP considers the information that a sample is distinguished from another sample as private. We want to humbly mention that [4] was published after the submission deadline of ICLR. Below we empirically compare our method (DP-OPT) to DLN-1 on privatized data [2]. We use Vicuna-7b to generate prompts and the same hyperparameters in Section 5.1. We show that our method can outperform [2] on sst2, trec, and mpqa significantly.
> >
> > Authors are confusing threat model and DP.
> >
> > (1) Multiple threat model can be used to test usefulness of any of DP models. DP definition and threat models are completely orthogonal.
> >
> > (2) LDP is stronger than DP. With LDP readily implying DP.
> >
> > (3) " ..... We consider the information of a sample, that can be recognized in a dataset, as private. In contrast, LDP or metric DP considers the information that a sample is distinguished from another sample as private...... " Indeed both are  different but one implies other.  Here is way to see it practically
> >
> > Let  $T$ be certain text and $s$ be its class label, then let $T' = (T,s)$. Then run unsupervised privitization mechanisms [1,2,4] on  $T'$. Example: Let $T$ be review and $s$ be its sentiment.  New Document is T' = "Review: <T>. Sentiment is <S>."  Now run any of document sanitisation methods.
> >
> > Also, these methods as said are task-agnostic and simpler.
> >
> > (4) Indeed, you don't have to compare with any recent works like [4]! My point is that you ignored a complete line of work that is extremely relevant and well-known, which is **3-4** years old like [1]. Furthermore, I understand there is a time constraint to compare, which is the reason I said, these methods have to be discussed in the related work and ideally compared in experiments. Such a discussion gives a complete understanding of the proposed methods and puts them into perspective for the reader.
> >
> >
> > > We compare our method to [2]. We use MIA as a privacy metric since their eps cannot be transformed to general DP. We also evaluate loss-based and Likelihood Ratio MIA attack (Mireshghallah, 2022) using the initial instruction as a reference model. However, we do not observe an obvious trade-off using MIA. For DP-OPT the MIA AUC is always around 50%.
> >
> > Thanks for these experiments.

---

> > > ### Author Response · Authors · 2023-11-21
> > > **Thank you for the follow-up questions**
> > >
> > > We would like to thank the reviewer for the prompt and detailed response! We agree that both LDP and DP can be adopted in our scenario. According to your suggestion, we have updated our related work section and included the discussion for these relevant works. In addition, we have included the comparison experiment from our rebuttal in Appendix B.2.
> > >
> > > For ease of reading, we also put the added discussion in the related work section below:
> > >
> > > “Another focal point among the community is the sanitization of texts (Feyisetan et al., 2020; Xu et al., 2020; Carvalho et al., 2023; Du et al., 2023; Utpala et al., 2023). These works introduce randomness at the word level and are able to achieve the privacy guarantee in terms of local differential privacy (LDP) (Kasiviswanathan et al., 2011) or a similar definition called metric differential privacy. Recent advancements in this field (Mattern et al., 2022; Utpala et al., 2023) combine the idea of perturbation with paraphrasing based on fine-tuning or zero-shot prompts.”
> > >
> > > We want to thank the reviewer again for pointing out this line of relevant works! If the reviewer has any additional comments or suggestions, we are more than happy to further revise our manuscript.

---

> > > > ### Comment · Reviewer_W4ZT · 2023-11-21
> > > > **Thanks for the reply**
> > > >
> > > > Thanks for modifications. I have now increased my score.

---

> > > > > ### Author Response · Authors · 2023-11-21
> > > > > **Thank you for raising the score**
> > > > >
> > > > > We are very grateful for your responsive replies which are very valuable for improving the paper. Many thanks for raising the score.

---

> ### Comment · Reviewer_W4ZT · 2023-11-21
> **clear procedure for private prompt tuning**
>
> For better clarity, I am explicitly listing the procedure that uses text sanitization methods
> 1) Utilizing clean documents to generate sanitized text documents.
> 2) Employing non-private discrete prompt tuning on these already privatized documents. This should maintain privacy due to the composition rule, assuming the label is not compromising.
> 3) Applying the learned prompt at inference time.
>
> Since the prompt is acquired from sanitized text documents, any potential data leakage would occur from paraphrases rather than the original content. I hope this clarifies better.

---

> ### Author Response · Authors · 2023-11-21
> **Update on private prompt tuning with text sanitization**
>
> Thank you for the detailed clarification! Accordingly, we updated our draft by enriching the implementation details in Appendix B.2. Our implementation follows exactly what you described.
>
> The updated content is marked as blue in Appendix, and we also put the description here for your reference.
>
> In Table 9, we compare our method to DLN-1 using sanitized data (Mattern, 2022), denoted as Private DLN-1. The implementation includes three steps:
> **(1)** First, the embedding of each token will be perturbed and projected into the original embedding space. This step can be extended to other sanitization methods like (Utpala, 2023, Feyisetan, 2020).
> **(2)** We use DLN-1 to tune prompts on these samples. DLN-1 is selected here due to its similarity to our method but can be replaced by other prompt-tuning algorithms in practice.
> **(3)** We use the generated prompts for inference.
>
> We want to thank the reviewer again for the suggestion! If the reviewer has any additional comments, we will further revise our manuscript!

---

### Meta-Review · Area_Chair_P2Rg · 2023-12-05

**Metareview:**

This paper introduces Differentially-Private Offsite Prompt Tuning (DP-OPT) as a solution for data privacy concerns when utilizing Large Language Models (LLMs). DP-OPT operates on the client side and offers an end-to-end framework to generate private and transferable prompts for cloud-hosted LLMs. It ensures data confidentiality, information privacy, and model ownership protection. The paper demonstrates that prompts tuned by LLMs can be effectively transferred across models and introduces a novel differentially-private mechanism for generating private prompts.

The vision and novelty are clearly above the bar of ICLR. For example, this paper solves a very important yet larger overlooked problem, i.e., the two-fold concerns surrounding data privacy when adapting LLMs on sensitive data. DP-OPT is the first end-to-end framework, where the entire prompt process is managed on the local device and offers services via an API, thus ensuring data confidentiality, information privacy, and cloud model ownership and IP. While the reviewers had some concerns on the experiments, the authors did a particularly good job in their rebuttal. For instance, they have included additional content regarding the real-world consequences of privacy leakage from prompts, according to the reviewer's suggestion. Therefore, all of us have agreed to accept this paper for publication! Please include the additional experimental results and discussion in the next version.

**Justification For Why Not Higher Score:**

The vision and novelty are clearly above the bar of ICLR. For example, this paper solves a very important yet larger overlooked problem, i.e., the two-fold concerns surrounding data privacy when adapting LLMs on sensitive data. DP-OPT is the first end-to-end framework, where the entire prompt process is managed on the local device and offers services via an API, thus ensuring data confidentiality, information privacy, and cloud model ownership and IP. While the reviewers had some concerns on the experiments, the authors did a particularly good job in their rebuttal. For instance, they have included additional content regarding the real-world consequences of privacy leakage from prompts, according to the reviewer's suggestion. Therefore, all of us have agreed to accept this paper for publication! Please include the additional experimental results and discussion in the next version.

**Justification For Why Not Lower Score:**

The vision and novelty are clearly above the bar of ICLR. For example, this paper solves a very important yet larger overlooked problem, i.e., the two-fold concerns surrounding data privacy when adapting LLMs on sensitive data. DP-OPT is the first end-to-end framework, where the entire prompt process is managed on the local device and offers services via an API, thus ensuring data confidentiality, information privacy, and cloud model ownership and IP. While the reviewers had some concerns on the experiments, the authors did a particularly good job in their rebuttal. For instance, they have included additional content regarding the real-world consequences of privacy leakage from prompts, according to the reviewer's suggestion. Therefore, all of us have agreed to accept this paper for publication! Please include the additional experimental results and discussion in the next version.

---

### Decision · Program_Chairs · 2024-01-16

Accept (spotlight)